# Where's the Rock: Using Convolutional Neural Networks to Improve Land Cover Classification

**Helen Petliak [1],\* , Corina Cerovski-Darriau [2] , Vadim Zaliva [3] and Jonathan Stock [2]**

[1]  Digamma.ai, 14500 Big Basin Way, Suite G, Saratoga, CA 95070, USA
[2]  U.S. Geological Survey, Menlo Park, CA 94025, USA; ccerovski-darriau@usgs.gov (C.C.-D.); jstock@usgs.gov (J.S.)
[3]  Carnegie Mellon University, NASA Research Park, Moffett Field, CA 94035, USA; vzaliva@cmu.edu
\*  Correspondence: lpetlyak@codeminders.com

**Abstract:** While machine learning techniques have been increasingly applied to land cover classification problems, these techniques have not focused on separating exposed bare rock from soil covered areas. Therefore, we built a convolutional neural network (CNN) to differentiate exposed bare rock (*rock*) from soil cover (*other*). We made a training dataset by mapping exposed rock at eight test sites across the Sierra Nevada Mountains (California, USA) using USDA's 0.6 m National Aerial Inventory Program (NAIP) orthoimagery. These areas were then used to train and test the CNN. The resulting machine learning approach classifies bare rock in NAIP orthoimagery with a 0.95 $F_1$ score. Comparatively, the classical OBIA approach gives only a 0.84 $F_1$ score. This is an improvement over existing land cover maps, which underestimate rock by almost 90%. The resulting CNN approach is likely scalable but dependent on high-quality imagery and high-performance algorithms using representative training sets informed by expert mapping. As image quality and quantity continue to increase globally, machine learning models that incorporate high-quality training data informed by geologic, topographic, or other topical maps may be applied to more effectively identify exposed rock in large image collections.

**Keywords:** remote sensing; environment; geology; land cover; land use; classification

## 1. Introduction

An increasing abundance of publicly-available Earth surface imagery makes it easy to find orthorectified imagery for nearly every location in the world, often at meter-scale or better resolution. Interpreting that imagery is another matter. For much of the past century, skilled experts used 1:20,000 or coarser (larger resolution) air photos to map regional interpretive maps showing topography, geology, ecology, water, soil, and many other features. In particular, maps of land cover are used to inform models that calculate carbon (and other nutrient) storage (e.g., [1] and references therein), water storage (e.g., [2,3] and references therein), the susceptibility to erosion (e.g., [4,5]) or mass movements like landslides (e.g., [6–9] and references therein). For all these examples, the calculations would vary depending on whether barren areas are bare rock or exposed soil. However, this is typically impossible with existing land cover maps. In the United States, Natural Resources Conservation Service (NRCS) staff (formerly the Soil Conservation Service) has mapped soil types across the country for agricultural purposes since 1899 and has used aerial imagery and nationwide databases since 1952 (https://www.nrcs.usda.gov/) [10]. These surveys are immensely useful to show the spatial distribution of soil and are a driving input for many soil and vegetation models (e.g., RUSLE2 for soil erosion [4,5]; LUCAS for carbon cycling [11,12]; LPJ and CLM-DGVM for dynamic vegetation models [2,3], however these surveys are typically based on ~10–100 m orthoimagery and

accurate mostly for flat, agricultural areas [13,14]. With the advent of satellite imagery (Landsat), a consortium of federal agencies started maintaining the National Land Cover Database (NLCD) (https://www.mrlc.gov/), which classifies the 30 m resolution satellite data into surface cover categories, like grasslands, wetlands, barren land, etc. (e.g., [15–17] and references therein). These thematic categories are broad and focused on vegetation cover, so "barren land" includes soil cover and rock without distinction. Overlaying an NRCS soil map, or a more evolved NLCD land cover map, on a Landsat image illustrates the opportunity to improve such maps. Figure 1 shows that existing maps tend to over- or underestimate the extent of rock and soil compared to more detailed mapping. Here, the 2001–2011 NLCD map has an overall accuracy of 0.53 and underestimates the rock by 88%, while the NRCS map has an overall accuracy of 0.75 and overestimates rock outcrops by 41%. This limits the utility for finer scale models.

The arrival of global, sub-meter-scale imagery in the last decade presents an opportunity to improve the resolution and extent of surface cover maps–a general term for the material exposed at the Earth's surface (e.g., rock, soil, water, vegetation). To improve these maps, one option would be to manually map rock outcrops from these new high-resolution imagery assets using expert knowledge. This kind of interpretive map could arguably be the most accurate and most useful, but it would also vary among individuals and be prohibitively time-consuming over a large area (e.g., [18,19]). Such a technique limits mappers to creating detailed surface cover maps of only small areas (Figure 1). Alternately, if the visual characteristics that a human expert sees could be captured and automated by training a computer that would negate those challenges. Automated methods are an increasingly common approach to classifying and making land cover maps (e.g., [20–22]). However, successfully training an algorithm is challenging because soils, geology, and ecology vary greatly by region. This variability defeats any global parameterization for surface cover classification. Additionally, soil and rock often have similar spectral characteristics, making training an algorithm particularly difficult. Therefore, to be successful, we propose machine learning techniques that require iterative expert mapping and programming input.

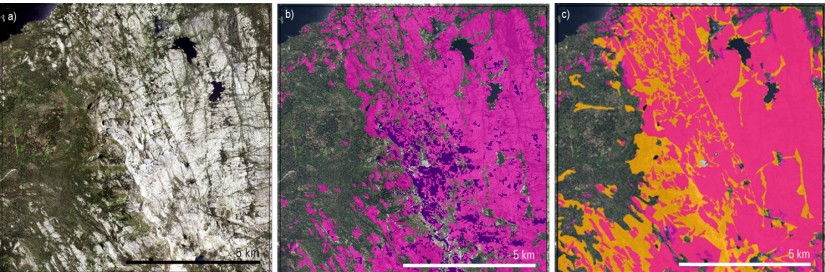

**Figure 1.** Comparison of hand-mapped bedrock outcrops (pink) to existing databases. (**a**) Location in the Sierra Nevada Mountains with exposed granite (bright areas), (**b**) showing bedrock is underestimated using the 2001–2011 NLCD "barren" class (purple), and (**c**) overestimated using NRCS soil survey units that are classified as >=50% "rock outcrops" (orange). This comparison illustrates that much improvement could be made to existing soil and land cover maps if remote sensing-based manual mapping could be scaled-up.

This paper explores how to improve surface cover classification by simplifying the problem to just differentiating exposed bedrock (*rock*) from bare soil (*other*) using machine learning. The accuracy of various machine learning approaches is tested using training data that separates exposed bedrock from other surface covers using USDA's publicly available National Aerial Inventory Program (NAIP) 0.6-m orthoimagery. Our approach is to:

- Use geologic maps and orthoimagery to guide test area selection
- Map rock versus other land surface covers at eight test sites in the Sierra Nevada Mountains, USA
- Develop a machine learning approach optimized to distinguish *rock* from *other* based on the training data

A geologic map is used to delimit areas of roughly equivalent reflectance properties (similar rock types) while orthoimagery is used to ensure test areas capture the range of spectral diversity (some variability in rocks, vegetation, etc.), thus improving the likely performance of machine learning models. Subsequent expert mapping within each test area provides an accurate initial map of bedrock exposed at the surface for algorithm training. Then, various image classification algorithms are tested to see which machine learning models most efficiently and most accurately differentiate rock from other surface covers. The result is an improved map of soil vs. rock cover that depends on expert mapping at local scales and captures the natural variability of geology at regional scales. Coupling satellite imagery, high-quality maps, and expert-guided mapping with algorithm development and the emerging field of machine learning can improve our ability to interpret large areas of the Earth's surface.

## 2. Related Work

### 2.1. Data and Limitations

The algorithm and the specific approach chosen must be tuned to the data used and the desired output. The most common data used for land cover classification are: spectral (e.g., imagery–satellite or airborne), topographic (e.g., elevation maps and their derivatives), spectral-temporal (e.g., repeat imagery), and high spatial or spectral resolution data (e.g., lidar or hyperspectral) [23]. Previous researchers have demonstrated different successful approaches for land cover classification using multi- or hyperspectral imagery [24], digital elevation models (DEM) [25], digital surface models (DSM) [22], and their derivatives. The classification accuracy, especially for vegetation, can be improved by combining NDVI (a normalized difference between visible and near-infrared bands) and DEM or spectral reflectivity from infrared bands [26]. When this approach is applied to time-series data, like Landsat 8 [27], it has even more discriminatory power (e.g., LCMAP https://www.usgs.gov/lcmap) [28]. The above data sets illustrate the potential for multi- or hyperspectral images to improve global land classification models. There are, however, limitations due to the large pixel size or limited spatial coverage. While Landsat 8 has a unique global coverage through time, the 15–100 m resolution limits improvement to existing land cover maps due to sub-pixel mixing (i.e., multiple classes in a single pixel) [29]. Other multi- or hyperspectral platforms have even larger pixels (e.g., MODIS) or lack widespread, cloud-free coverage (e.g., ASTER). Lack of coverage or lack of high-resolution data are challenges for any workflow that seeks to improve surface cover maps in steep areas around the world.

Therefore, we explore the utility of using wider coverage, sub-meter-scale aerial imagery from USDA's 2016 NAIP collection [30]. This 4-band (R, G, B, NIR), publicly available (https://earthexplorer.usgs.gov/) imagery covers all of our test areas with cloud-free, 0.6 m pixels. The high-quality orthorectification process results in horizontal accuracy $+/-4$ m, an improvement over most widely available orbital imagery. With this imagery, bedrock outcrops (on the order of 10 m) can be visually mapped, and the infrared band provides an additional channel for model tuning. With availability for all of California, this workflow could potentially be expanded statewide in the future, or at least across the entire Sierra Nevada range. The workflow outlined in this paper could be applicable anywhere that NAIP or similar high-spatial resolution (i.e., meter to sub-meter), well-orthorectified data exists. The several-meter positional accuracy also allows for incorporating time series or other rectified data.

### 2.2. Potential Classification Models

Past land cover classification approaches use parametric (e.g., maximum likelihood or Bayesian classifiers) and non-parametric (e.g., Support Vector Machine, Random Forest, or Artificial Neural Network) classifiers [31]. These classical approaches depend on how scientists ultimately want to classify the categories: pixel by pixel [23,26,32]; sub-dividing each pixel using unmixing models (e.g., Fuzzy Approach [33,34]; Spectral Mixture Analysis [35,36]); or as objects (Object-Based Image

Analysis (OBIA) [37,38]). These methods are based on hand-crafted features [39], which are extracted from images according to a fixed, manually defined algorithm based on expert knowledge (e.g color, combination of spatial and spectral information).

Recent advances in image-based classification instead build on convolutional neural networks (CNNs). A CNN consists of a convolutional layer, non-linear mapping, pooling layer, and then an output layer that generates predictions. Deep CNNs consist of a series of such layers followed by a fully connected (FC) layer to stack all extracted features from the previous processes to generate an overall prediction [40]. Using deeper networks, such as VGG-16 and VGG-19 [41], may further increase classification accuracy, but each additional layer increases calculation time as it adds many more parameters. GoogLeNet [42] is an example of a Deep CNN approach that uses more but smaller kernels in the convolutional layers, so the number of parameters is smaller than VGG-16 despite increased depth. It also omits FC-layers to further reduce the number of parameters. ResNet [43] offers another architecture that may have more than 100 layers. This is achieved by shortcut connections that bypass some convolutional layers, so that only a residual function needs to be learned by the network. There are two main representatives of deep learning networks: Fully Convolutional Networks (FCN) and patch-based CNN. CNN and FCN tend to achieve similar accuracy, but FCN is more efficient in utilizing surrounding label information and in memory usage [44,45].

Previous studies have used CNNs to classify imagery by land cover. For example, Al-Najjar et al. [22] used pixel-based combination of four CNNs to classify 0.5 m aerial imagery with the aid of a digital surface model (DSM) into five land cover classes (vegetation, ground, road, building, and water). This model achieved a classification accuracy of 0.945. In another work [20], CNN was integrated with pixel-based multilayer perceptron (MLP) using a rule-based decision fusion strategy to classify the WorldView-2 satellite sensor dataset. This model achieved an overall classification accuracy of 0.87–0.89. Maltezos and Doulamis [46] implemented CNN for extracting buildings orthoimages using the height information as an additional feature. Pan and Zhao [47] proposed a classification method based on CNN and the restricted conditional random field algorithm (CNN-RCRF) to classify land cover into six classes (impervious surfaces, buildings, low vegetation, trees, cars, and clutter/background) using high-resolution remote sensing images. This method was used to avoid boundary distortions of the land cover and reduce computation time in classifying images. Their final model achieved an overall accuracy of 0.82.

The above studies show that classical models based on hand-created features are easily interpreted and assume that the extracted features are robust to the variances in the training data but need to be manually engineered [48]. Deep learning-based models, like CNN, present a generalized approach using feature extraction and classification in one trainable model. However, analyzing and visualizing the features is difficult [49]. The variety of methods used in recent studies shows there is no consensus yet on the best model for solving land cover classification problems. Here, we focus on the successes of previous CNN approaches but build a generalized model that improves the accuracy and specifically differentiates between soil and rock. We will also compare our results to several classical approaches based on hand-crafted features: Support Vector Machine (SVM), Random Forest (RF), Object-Based Image Analysis (OBIA). We test the accuracy of each approach by comparing our results to the eight expert mapped sites. Additionally, we aim to build a CNN architecture without any ancillary data in contrast to the deep-learning classification methods mentioned above, where the highest accuracy models require ancillary data, such as high-resolution DSM that is not widely available.

## 3. Methods

### 3.1. Data Sites

To map soil and bedrock, the model was trained in the Sierra Nevada mountains of California, USA (hereafter–Sierras) (Figure 2). The Sierras are an ideal test area because glaciers have exposed large swaths of bedrock, and the high-albedo granite contrasts with the darker vegetation, highlighting many

soil/bedrock boundaries. We focus on the central Sierras where the granite dominates, but localized areas of volcanic and older sedimentary and metamorphic rock also exist. This provides bare bedrock that is both easily visible and identifiable as well as some diversity of rock types and land cover for model training.

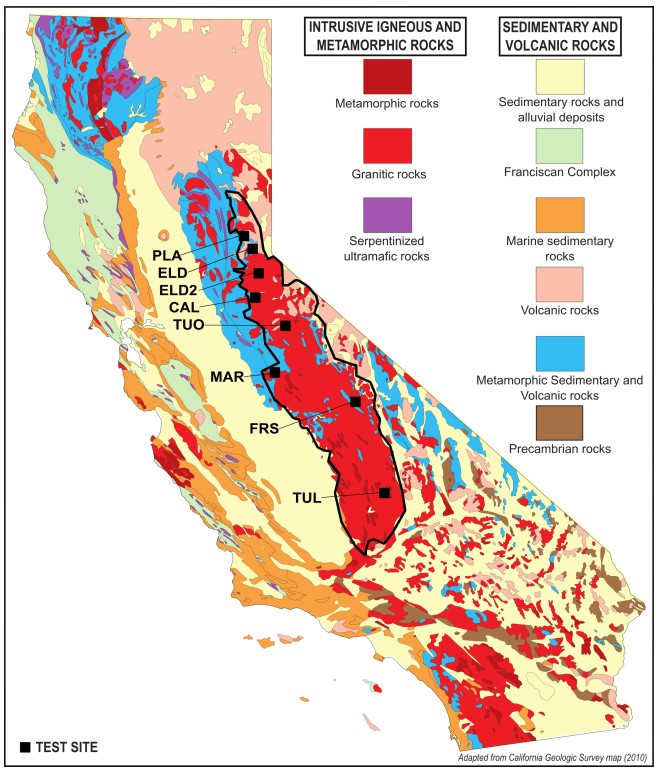

**Figure 2.** Simplified geologic map of California (adapted from [50]) with test sites marked (black squares). The model was trained and tested in the predominately granitic rocks (red) of the Sierra Nevada Mountains (black outline).

Building a surface cover classification model using remote sensing imagery of heterogeneous landscapes is challenging because spectral similarity between different classes (i.e., urban areas and bare rock, agricultural land and naturally bare soil) are common. Accurate classification of such landscapes is difficult and relies on sufficient, high-quality labels in the training data. Therefore, eight test sites from across the central Sierras were chosen to span the variety of bedrock types (e.g., granite to metamorphic), vegetation (e.g., grass to dense forest), textures (e.g., fine sands, gravel deposits, and boulders in streams), and colors (e.g., light to dark) known to exist in the area (Figure 2). Figure 3 shows the distribution and spectral variety of the 8 test sites, named by county: CAL, FRS, MAR, PLA, ELD, ELD2, TUL, TUO. Table 1 gives a brief description of the key features of each test site.

**Table 1.** Test site descriptions.

| Site Name | County | Location (NAD83) | Key Features |
|---|---|---|---|
| CAL | Calaveras | (38.43, −120.31) | Trees (logged areas and dense forest) |
| ELD | El Dorado | (38.98, −120.25) | Mostly light bedrock, trees |
| ELD2 | El Dorado | (38.69, −120.18) | Dark bedrock, trees, bare areas |
| FRS | Fresno | (37.10, −118.64) | Gravel, water, snow |
| MAR | Mariposa | (37.49, −119.97) | Urban area, trees, grass (mixed–brown and green) |
| PLA | Placer | (39.22, −120.46) | Dark and light bedrock (in place rock), gravel (loose rock), trees |
| TUL | Tulare | (35.92, −118.18) | Light bedrock, grass (brown) |
| TUO | Tuolumne | (38.09, −119.78) | Light bedrock, grass (brown) |

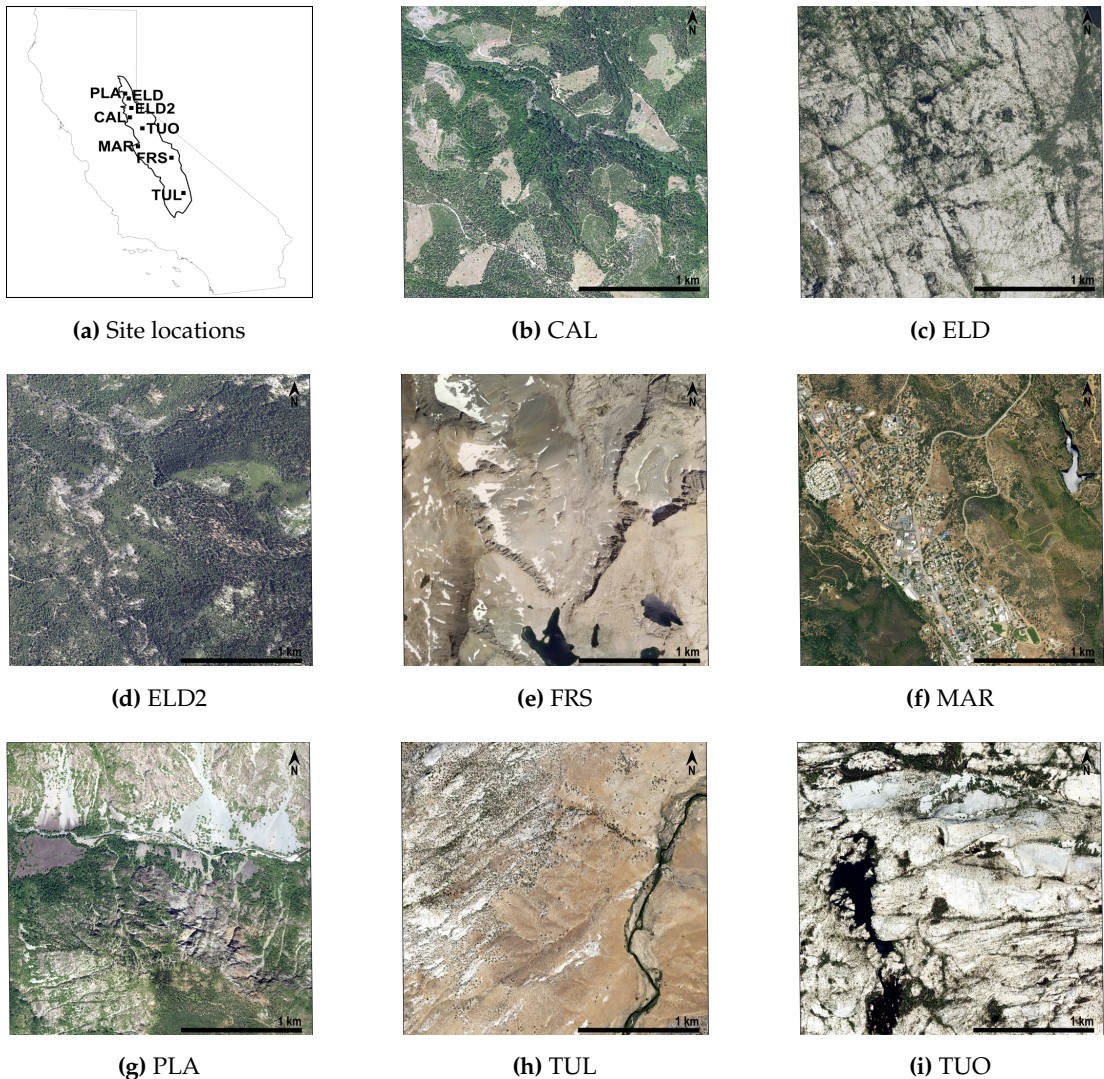

**Figure 3.** Locations of each 2.5 × 2.5 km test site (**a**) and 2016 NAIP Imagery used for each site (**b–i**).

At each site, a 2.5 km × 2.5 km area was mapped using a combination of semi-automated and manual mapping techniques to divide the surface into two classes: *rock* and *other*. We found that having only two categories led to falsely categorizing soil as bedrock and vice versa. Therefore, we first classified each test area as one of nine categories: bare rock, soil, gravel, grass, vegetation (e.g., trees and shrubs), roads, urban areas, snow, and water, using a python-based interactive classification tool, called "create_groundtruth," developed by Buscombe and Ritchie [51] (https://github.com/dbuscombe-usgs/dl_tools). Splitting the classification into nine categories greatly improves the accuracy of site labeling and thus improves the final model accuracy. Next, the classifications were manually verified using Google Earth, the 2016 NAIP imagery, and field checking.

In order to confirm that the pixels were correctly labeled, rocks were visually identified by texture (e.g., more relief than adjacent gravel or soil), by color (e.g., dark, lichen-covered rocks to bright, recent exposures), and by outcrop pattern (e.g., fractured and rounded outcrops typical of glaciated granite), often checking from multiple images and angles. To help identify ambiguous features, known rock outcrops were calibrated for comparison. Between 1–7% of the pixels had to be manually corrected during this verification process. The final "bare rock" polygons outlined exposed bedrock outcrops that are in-place and devoid of soil cover. They did not include loose or mobile rock (e.g., river or talus deposits), although sub-meter shrubs or trees are often included. Lastly, for each test area, the

categories were simplified to a binary mask (*bare rock* vs. *other*) to speed up training and running of the various test models.

Despite this effort to create accurate, expert-checked training data, the heterogeneous nature of rock surfaces, the inclusion of single trees or shrubs in the bare rock polygons, shadows, and human error all introduce noise that make developing a highly accurate machine learning model challenging (Figure 4). Therefore, we iterated through the test areas using the following workflow to improve the accuracy of the training data and develop a testing subset to check the accuracy of the final machine learning model.

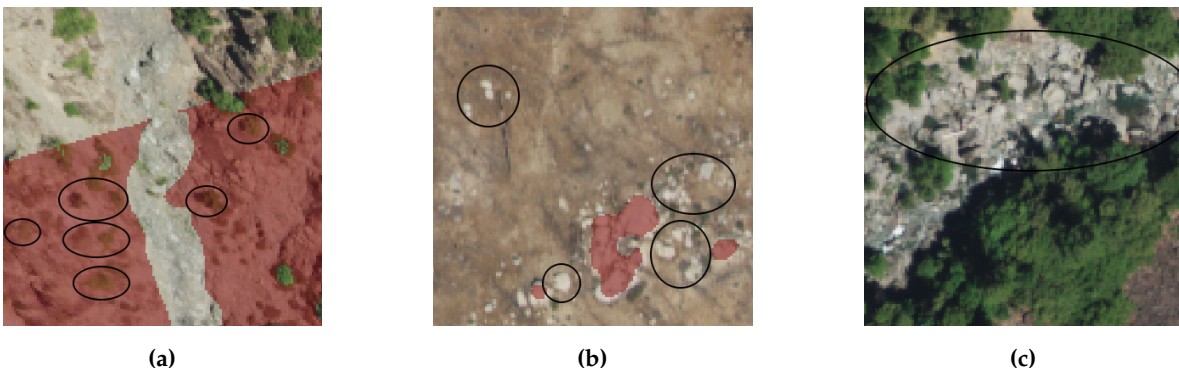

| **(a)** | **(b)** | **(c)** |

**Figure 4.** Illustration of false classifications on initial training data. In each scene, the red areas were initially labeled as *rock*. (**a**) Circled areas are false positives where trees were labeled as *rock*; (**b**) The circled areas show false negatives where high-albedo rock was labeled as *other*; (**c**) The circled area is ambiguous and may be bedrock or boulders in a stream.

## *3.2. Workflow*

Figure 5 illustrates our workflow, starting at the upper right, with the original NAIP imagery. A combination of manual and semi-automated mapping approaches were used to classify the initial data, which was then refined using manual and automated techniques. The initial mapped data is pre-processed and split into a training (65%) and a testing (35%) dataset. The first was used to train the models, which were then run on the original NAIP imagery. The resulting predicted model values were compared with the original ground truth labels from the testing subset to evaluate the accuracy. This allowed exploration of a variety of machine learning models in order to select the best match for the original mapped test site data.

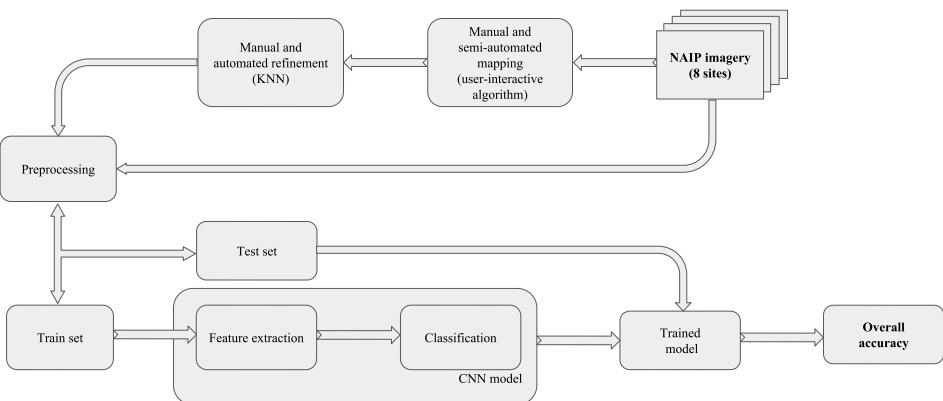

**Figure 5.** Workflow diagram for imagery classification and model development.

### 3.3. Label Refinement Using K-Nearest Neighbors (KNN)-Matting

The model accuracy depends on the quality of the training data label. Therefore, it is necessary to remove as many false positive and false negative errors from the original ground truth labels as possible. Manual reviewing of high-resolution data requires a lot of resources so instead, an image matting algorithm was used to improve the label quality as it needs only a relatively small subset of pixels.

Image matting is a common technique used for image segmentation and here, K-Nearest Neighbors (KNN) matting is used because it is specifically designed for the problem of extracting image layers simultaneously with sparse manual labels [52]. To refine the data, the KNN algorithm operates on a user-supplied classification, or mask, that includes three categories: *rock* (1), *other* (0), and undefined pixels.

This mask can be sparse where most of the pixels are unmarked. The algorithm then propagates labels from this initial mask to the undefined pixels, producing high-detail ground-truth labels that reduce noise and can then be used for model training. The steps in the KNN-process are shown in Figure 6, which illustrates that only a small subset of the input test area needs to be marked up to improve the label quality.

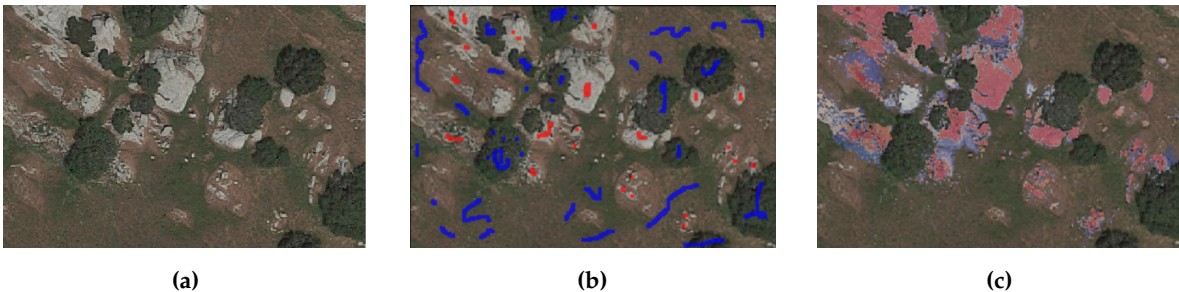

| (a) | (b) | (c) |

**Figure 6.** Illustration of KNN-process for label refinement. (**a**) NAIP imagery from ELD2; (**b**) sparse input mapping for KNN process, showing a subset of bare rock pixels classified in red and other pixel examples marked in blue; (**c**) KNN propagated final label classifications, with *bare rock* in red and *other* shown as transparent for clarity.

Two additional processing steps were performed prior to KNN-matting for two sites (ELD2 and CAL) with large amounts of visual noise (shadows, trees on rocks, etc.): (1) initial label erosion to remove errors on the edges, and (2) adding rough labels to fix false positive and false negative errors. This additional pre-processing and KNN-matting resulted in 15% (ELD2) and 11% (CAL) of pixels changing value from the initially mapped values. The result is shown in Figure 7. The KNN-matting improves the mask boundaries and excludes pixels that include trees or grass. These KNN-refined labels are subsequently used for ELD2 and CAL in place of the initial labels.

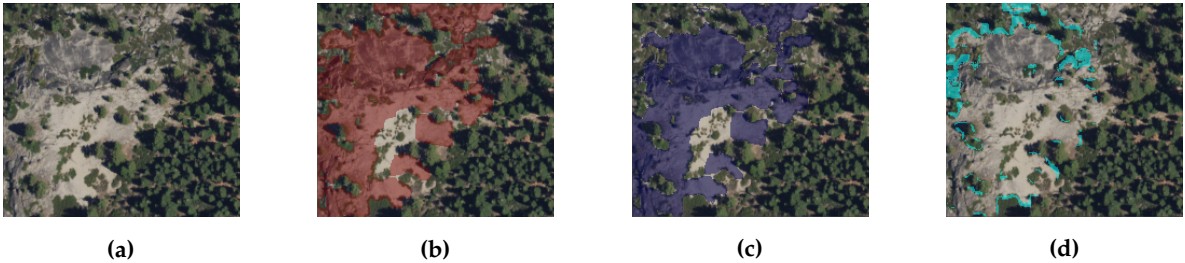

| (a) | (b) | (c) | (d) |

**Figure 7.** Comparison of initial labels to KNN-improved labels. (**a**) NAIP imagery from ELD2 tile; (**b**) initial bedrock mapping in red; (**c**) KNN-refined label in blue; (**d**) difference between initial and KNN labels in light blue.

### 3.4. Additional Pre-Processing

CNN only predicts one label per image (e.g., AlexNet [53]), so it cannot be applied directly to obtain a prediction on a per-pixel level as is required for land cover classification. One way to solve this problem is to use a moving window approach that splits the data into small pieces (windows) with a 1-pixel shift to perform per-pixel prediction (the number of windows is equal to the number of pixels in the image). The CNN label for each resulting window is then representative of its central pixel, *P* (Figure 8). The pixels surrounding the central pixel provide contextual information that is converted into a feature vector and used for training the model.

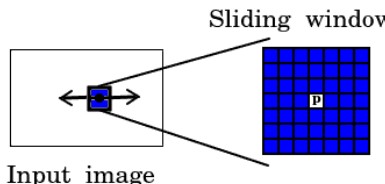

**Figure 8.** The sliding windows form the neighborhood of the pixel *P*.

To avoid any spatial correlation between the initial mapping and the model outcome, each of the eight mapped test areas was split into non-intersecting 300 × 300 pixel squares. These squares were divided randomly into the training (65%) and testing (35%) datasets with the same label proportions (i.e., the ratio of the two classes were equal in the training and test sets). Each 300 × 300 square was split into patches using the sliding window approach to perform per-pixel classifications, as shown in Figure 9. This procedure was repeated five times to perform cross-validation.

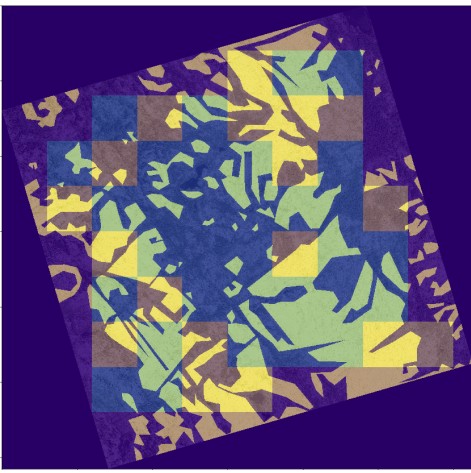

**Figure 9.** Example of training/testing splits for ELD: yellow squares–testing dataset (35%), blue rectangles–training dataset (65%).

### 3.5. Classical Models Based on Hand-Crafted Features

Two classical approaches (pixel-based, object-based) using hand-crafted features and SVM or Random Forest (RF) as classifiers [54] were compared to the CNN model. These classical approaches are computationally expensive because the number of windows required equals the number of pixels in the image, and existing classifier implementations require all data to be stored in memory. Therefore, to test these two approaches, we applied it to a subset of the two sites, ELD and ELD2, where bright granite bedrock exposures provide exceptional contrast.

To generate features, a set of multi-scale texture measures needs to be calculated for each of the 4 NAIP channels: mean, variance, kurtosis, and skewness. Additionally, the means and variances were extracted from Sobel, Laplacian, and Canny [55] low-level texture descriptors. Features of

the gray-level co-occurrence matrix [56] (correlation, dissimilarity, homogeneity, contrast, energy) were also used. Finally, a histogram was used with eight bins for each channel as a separate feature. These features were passed to the RF or SVM classifier. The RF classifier consisted of a combination of tree classifiers, where each classifier is generated using a random vector sampled from the input vector, and each tree casts a unit vote for the most popular class in order to classify the input vector [57]. The SVM classifier mapped input vectors non-linearly to high-dimension feature space and constructed a linear decision surface is this feature space [58].

Additionally, RF allows extraction of feature importance (Table 2). Since many features are generated but some are less informative, determining feature importance provides an understanding of which variables are the most predictive in these models. Comparing the feature hierarchy with expert knowledge helps assess the adequacy of the model. For example, the high importance of features associated with the infrared band (5–7%) could be explained by the spectral reflectance of rock being controlled, for the most part, by four variables: moisture content, iron oxide content, mineralogy, and structure. All of these characteristics are typically distinct from soil and vegetation cover and are associated with high infrared reflectivity.

**Table 2.** Random Forest (RF) feature importance.

| Name | Importance |
| :---: | :---: |
| Infrared pixels between 128 and 160 | 7% |
| Infrared pixels between 196 and 228 | 5% |
| Skewness of blue | 5% |
| Mean of red | 5% |
| Mean of green | 4% |
| GLCM correlation vertically | 3% |
| Skewness of red | 3% |
| GLCM homogeneity infrared | 3% |
| Blue between 196 and 228 | 2% |
| Laplacian of green | 2% |

We explored the use of object-based image analysis (OBIA) classifiers. Unlike our previous approach, where we used a patch to classify the central pixel, OBIA classifies a group of pixels (an object or a segment). A graph-based image segmentation algorithm (Figure 10) was used to merge pixels. From each segment, the features were extracted and used to train the model and predict one class label per segment using RF or SVM.

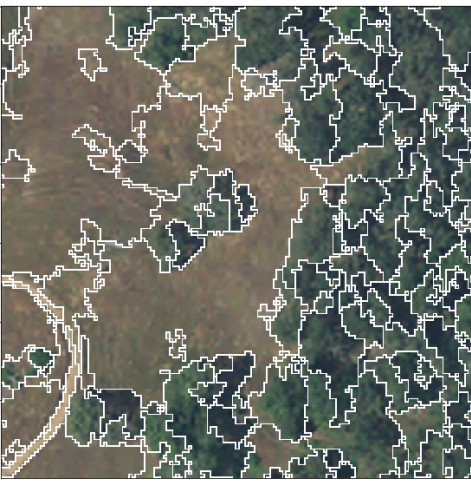

**Figure 10.** Object-based Image Analysis (OBIA) segmentation result for one of ELD's test rectangles, showing a grouping of similar pixels merged into objects separated by white boundary.

### 3.6. CNN Model

A more generalizable and robust option is a convolutional neural network (CNN). For both options, the quality of the classification depends mostly on the quality of the features extracted from the image. The classical approaches rely on manual extraction of hand-crafted features [59], whereas CNN relies on automatic extraction using deep learning architectures [60]. The latter has the ability to combine feature extraction and classification into one trainable model. The discriminative power of the learned features in CNN is much higher because the deep learning architecture is capable of learning both low-level features (edges, curves, lines, etc.) and more abstract features (objects, shapes, etc.) using a number of convolutional layers.

A convolutional layer is a set of different filters of a defined size (e.g., $3 \times 3$, like in VGG16) that detect straight edges, colors, curves, etc. Starting from the top left corner, the filter slides (convolving) over the input image, the result of each convolution operation is a single number. After sliding each filter over the image, the result is an activation map (or feature map). The more filters the bigger the depth of the activation map and thus the more information that can be extracted from the input image. In a traditional convolutional neural network architecture, there are additional layers that follow the convolution layers. In general, they simplify (by down sampling and summarizing) the feature maps to reduce non-linearities and dimensions (e.g., *Max-pool* layer), which help to improve network robustness and control for overfitting. The last layer (*Fully Connected* layer) stacks all extracted features from the previous layers into a vector that gives the probability distribution used to predict the label for the input image. Here, we built a nine-layer CNN to classify the likelihood of each pixel, $p$, being labeled *rock* by analyzing that central pixel and its neighbors. Figure 11 illustrates the architecture, starting at the left with inputs of 4-band NAIP patches. The first and the second convolutional layers consist of eight filters with a size of $3 \times 3$ and use zero padding to keep the spatial dimension of the resultant feature maps. The last convolutional layer consists of 16 filters. *Max-pooling* is always applied with a window of $2 \times 2$. For each pixel $p$ being classified, the output is a posterior probability (i.e., the conditional probability given the knowledge from the training data) for the pixel to be *rock*. Different input patch sizes ($15 \times 15$, $31 \times 31$, $65 \times 65$, and $97 \times 97$) were tested to determine which of them yields the highest classification accuracy. We added NDVI as the 5th band, but it did not improve the results, so we dropped this step from the workflow.

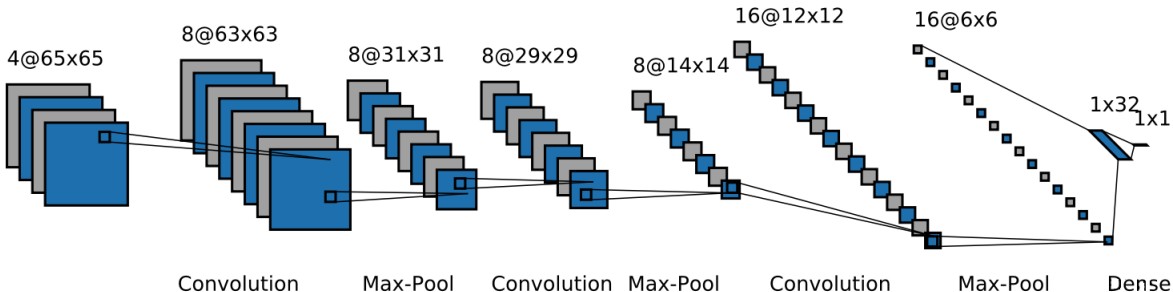

**Figure 11.** Nine-layer CNN architecture used to classify pixels as "rock".

Lastly, an Adam optimizer [61] with an experimentally selected learning rate of 0.001 was used to train the CNN model. Adam is an optimization algorithm for deep learning models that iteratively updates parameters based on training data. The algorithm functions by minimizing the gradient descent and uses binary cross-entropy loss to measure performance (Equation (1)),

$$L\left(\theta\right) = -\frac{1}{N}\sum_{i,k} y_{i,k} \log\left(\hat{y}_{i,k}\right) + \left(1 - y_{i,k}\right)\log\left(1 - \hat{y}_{i,k}\right) \qquad (1)$$

where $k$ is the index of an image, $N$ is the number of images in a mini-batch, $y_{i,k}$ is the ground truth label, and $\hat{y}_{i,k}$ is the predicted value. A *mini-batch* [62] implementation of gradient descent was also

used, which allowed us to split the training dataset into small batches used to calculate model error and update model coefficients. This obviated the need to store an entire data array for training models, reducing memory usage, and allowing us to train the final model using all eight areas. In order to compensate for an imbalanced class distribution of the training data, the batch was collected equally from the positive and negative classes.

All models were implemented based on the Tensorflow framework, and all experiments were executed on the Amazon EC2 *3.8xlarge* instance with 2 NVIDIA Tesla M60 GPUs, each with 2048 parallel processing cores and 8 GB of video memory and 32 High frequency Intel Xeon E5-2686 v4 (Broadwell) processors. CNNs were trained approximately two hours per epoch, and the full training took about 20 h.

*3.7. Accuracy Assessment*

The performance of our models was evaluated by calculating the Kappa statistic and $F1_{Macro}$. Kappa measures the agreement between multiple methods of categorical assessment (where values closer to 1 indicate better agreement), and is used to compare the various classical methods and the CNN model. $F1_{Macro}$ metrics, calculated as harmonic mean of Precision and Recall with macro-averaging, is used as the accuracy assessment since it is less affected by multiclass imbalance (i.e., more pixels were classified as *other* than as *rock*) [22]. Mathematically, $F1_{Macro}$ can be expressed as shown in Equations (2)–(4):

$$Precision_{Macro} = \frac{\sum_{i=1}^{N} \frac{TP_i}{TP_i + FP_i}}{N} \tag{2}$$

$$Recall_{Macro} = \frac{\sum_{i=1}^{N} \frac{TP_i}{TP_i + FN_i}}{N} \tag{3}$$

$$F1_{Macro} = 2 * \frac{Precision_{Macro} * Recall_{Macro}}{Precision_{Macro} + Recall_{Macro}} \tag{4}$$

where $TP_i$ is the true positive, $TN_i$ is the true negative, $FP_i$ is the false positive, $FN_i$ is the false negative for class-i, and $N$ is the number of classes. To assess the statistical difference between the classifiers, McNemar's test was used [63].

Due to the high class imbalance, another unambiguous way to present the prediction results of a classifier is to use a confusion matrix. For a binary classification, the matrix has two rows and two columns. Across the top of the confusion matrix are the predicted class label, and down the side are the observed class label. Each cell contains the fraction of predictions made by the classifier that fall into the true positive, true negative, false positive, or false negative categories.

## 4. Results

Table 3 shows the mean $F1_{Macro}$ score (over all eight sites) using input patch sizes from 15 to 97 pixels before and after refinement. A patch size of $65 \times 65$ yielded the highest overall $F_1$ score. The table also illustrates that using label-refined data for training a CNN model can improve classification accuracy by 0.04–0.06.

**Table 3.** $F_1$ score for Various Patch Sizes.

| Window Size | $F1_{Macro}$ (Before Label Refinement) | $F1_{Macro}$ (After Label Refinement) |
|---|---|---|
| 15 × 15 | 0.83 | 0.87 |
| 31 × 31 | 0.84 | 0.90 |
| 65 × 65 | 0.90 | 0.95 |
| 97 × 97 | 0.87 | 0.91 |

The accuracy of the pixel-based and object-based classification tests compared to the refined CNN model (65 × 65 window) is shown in Table 4.

**Table 4.** Model comparison.

| Model | Training Data | $F1_{Macro}$ | Kappa |
|---|---|---|---|
| RF classifier (pixel) | ELD, ELD2 | 0.78 | 0.78 |
| SVM classifier (pixel) | ELD, ELD2 | 0.80 | 0.79 |
| RF classifier (OBIA) | ELD, ELD2 | 0.82 | 0.81 |
| SVM classifier (OBIA) | ELD, ELD2 | 0.84 | 0.84 |
| Proposed model (CNN) | ELD, ELD2 | 0.93 | 0.92 |
| Proposed model (CNN) | Eight sites | 0.95 | 0.95 |

Each of these classical approaches have lower mean $F1_{Macro}$ and Kappa scores, which are measures of classification accuracy where values closer to 1 indicate increased accuracy. In addition, there were a number of difficulties encountered while implementing the classical approaches. The methods implemented for extracting spatial features generate only low-level features requiring empirical parameters (e.g., neighbor size). Spatial features, therefore, depend on expert knowledge and parameter setting, which is why it is difficult to find universal parameters to extract appropriate features for each type of land cover surface using these methods. Consequently, spatial features are usually not robust and have poor generalizability, leading to lower accuracy.

The CNN model had the highest $F1_{Macro}$ score, thus most accurately classified bare rock in this study even with the class imbalances. The final CNN model, trained over all eight sites using the refined labels, produced the most accurate bedrock (*rock*) vs. soil-cover (*other*) maps (0.95 $F1_{Macro}$ score) using high spatial resolution (0.6 m) 4-band NAIP imagery without the help of ancillary data, such as texture, slope, or elevation from DEM data. The McNemar test statistic is 15 (*p*-value < 0.001), meaning the CNN had a statistically significant improvement compared to SVM-classified OBIA (the most accurate classical test).

To calculate the classification accuracy for each of the 8 test sites, we calculated a separate confusion matrix for each site to identify which had the most true positives (*TP*)(bottom right) and true negatives (*TN*) (upper left) of each matrix in Table 5. We used Equations (2)–(4) to calculate the overall accuracy ($F1_{Macro}$) for each site. $F1_{Macro}$ was lowest for the CAL (0.93), PLA (0.92), and MAR (0.93) sites. This is probably due to the fact that CAL and MAR have patches of dry grass, which has a similar spectral reflectance as light-colored bedrock, so it is difficult to differentiate with the given NAIP spectral bands. Similarly, PLA has large talus piles (loose rock) that are spectrally similar to bedrock.

**Table 5.** Confusion Matrix and $F1_{Macro}$ for Each Test Site.

| | | Predicted Class Other | Rock | Site | Predicted Class Other | Rock | Site |
|---|---|---|---|---|---|---|---|
| **Observed class** | **other** | 0.97 | 0.03 | ELD2 | 0.96 | 0.04 | CAL |
| | **rock** | 0.04 | 0.96 | | 0.11 | 0.89 | |
| | $F1_{Macro}$ | 0.97 | | | 0.93 | | |
| **Observed class** | **other** | 0.96 | 0.04 | PLA | 0.94 | 0.06 | TUL |
| | **rock** | 0.12 | 0.88 | | 0.06 | 0.94 | |
| | $F1_{Macro}$ | 0.92 | | | 0.94 | | |
| **Observed class** | **other** | 0.97 | 0.03 | ELD | 0.97 | 0.03 | MAR |
| | **rock** | 0.04 | 0.96 | | 0.11 | 0.89 | |
| | $F1_{Macro}$ | 0.97 | | | 0.93 | | |
| **Observed class** | **other** | 0.98 | 0.02 | FRS | 0.97 | 0.03 | TUO |
| | **rock** | 0.09 | 0.91 | | 0.08 | 0.92 | |
| | $F1_{Macro}$ | 0.95 | | | 0.95 | | |

Examples of the prediction results from the CNN model compared to the initial mapping are shown in Figure 12. The CNN model does a cleaner job of separating trees from rocks, though the accuracy was lowest in the areas at the intersection of rock and non-rock surfaces.

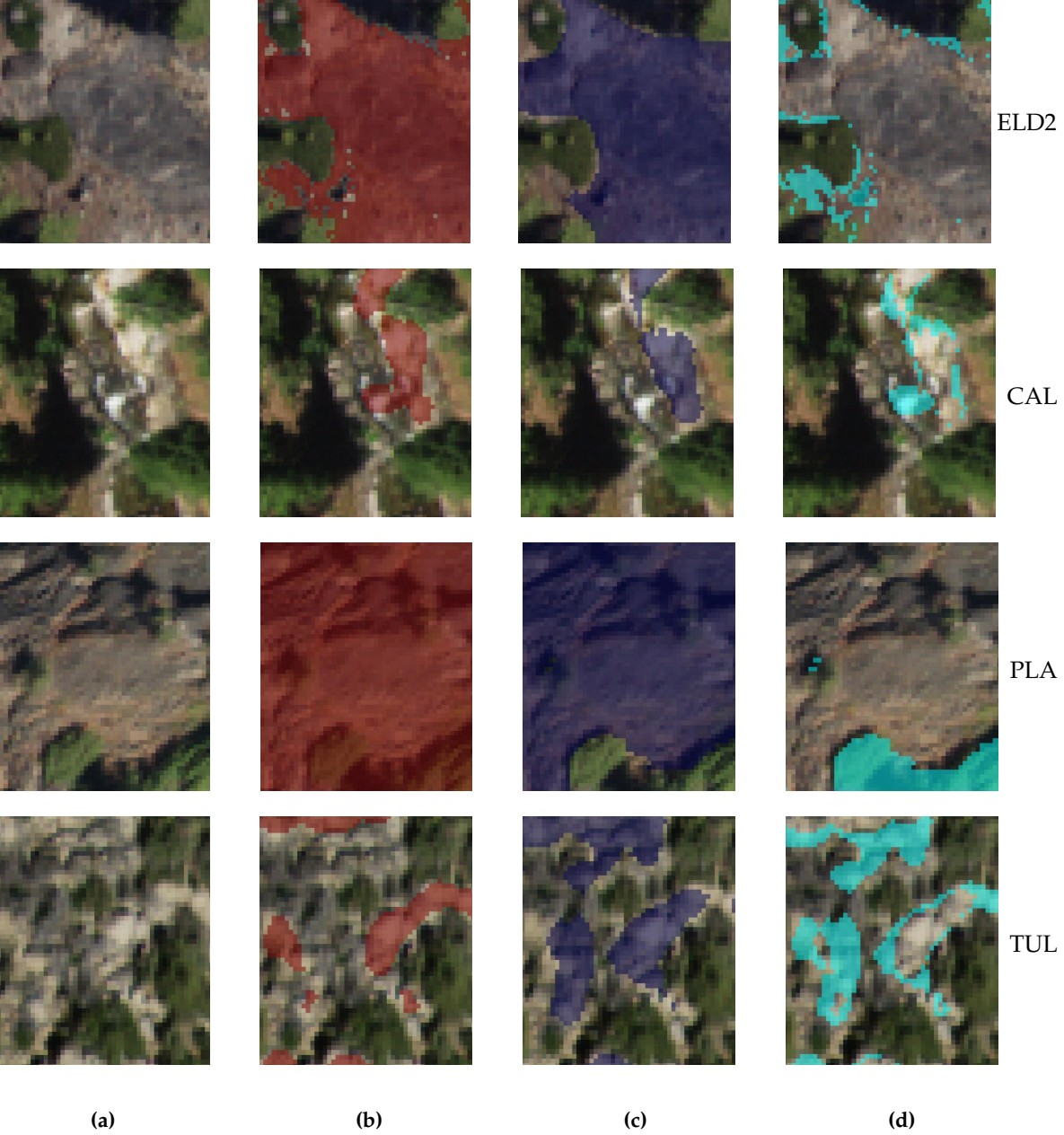

(a)          (b)          (c)          (d)

**Figure 12.** Sample results from 4 test sites: (**a**) input image; (**b**) initial mapping; (**c**) CNN prediction; (**d**) difference between prediction (**c**) and initial mapping (**b**).

Even with the more generalized CNN approach that combines feature extraction and classification, feature maps extracted from the first convolutional layer (Figure 13), in some cases, can still be interpreted, for example, as extracted low reflected areas (shadows) (Figure 13c), green areas (Figure 13d,i), and textures (Figure 13g,j,k). These maps provide a better understanding of which features are automatically learned during the training process.

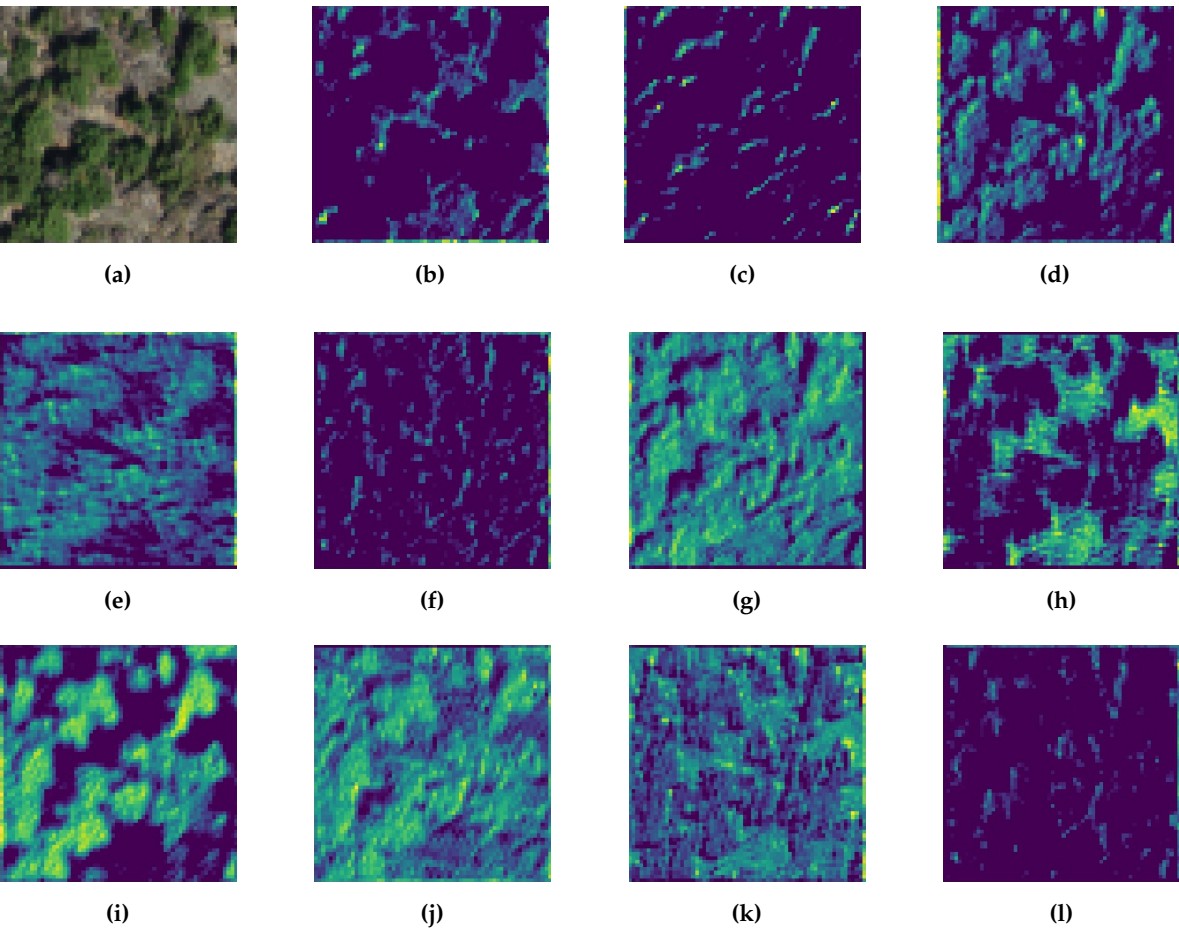

**Figure 13.** First convolutional layer features: (**a**) input image; (**b**,**h**) light areas; (**c**,**f**,**l**) low reflected areas (shadows); (**d**,**e**,**i**) green areas; (**g**,**j**,**k**) textures (roughness or smoothness).

## 5. Discussion

We focused on specifically improving rock classification, as it is currently lacking or combined into other classes in existing surface cover datasets (e.g., NLCD, NRCS, etc.). This paper presents a workflow that has identified rock outcrops with an overall 0.95 $F1_{Macro}$ for eight test sites, which is an improvement over the mere 0.53–0.75 overall accuracy of the existing datasets (e.g., Figure 1). This model, which specifically adds *rock* identification, performed as well if not better than those of previous similar CNN studies that had 0.82–0.95 overall accuracy rates [20,22,47]. Therefore, the method presented here offers a large improvement over what is currently available for barren or rock classification, even if it does not refine the class labeled *other*. Distinguishing between *rock* and *other* is still useful, especially in natural (non-urban) environments (like the Sierras). In natural settings like the Sierras, *other* can be a useful proxy for the presence of soil (as opposed to exposed bedrock). This can be used for resource, erosion, or hazard modeling. For example, hazardous shallow, rainfall-induced landslides only occur in soil. Therefore, knowing where soil is present (as opposed to rock) helps delineate where this kind of landslide is most likely to occur. Having a better surface map of rock exposure will limit the spatial extent over which landslide susceptibility models need to be run for a region of interest, making the resulting model more accurate and computationally less time intensive.

The CNN workflow requires accurate mapping of *rock* vs. *other* in the initial training dataset. However, this workflow adds corrections for "noisy" initial mapping. This means that human bias in mapping or imprecise initial mapping is less problematic. Thealgorithm for improving the labeling process, based on KNN-matting, significantly decreases the number of resources needed to perform

high-quality corrections. This means that training data can be mapped more quickly and include some mixed pixels (e.g., trees on rocks).

The model developed here illustrates a simpler, more general approach by combining feature extraction and classification in one trainable model without additional data beyond the 4-bands NAIP imagery. The highest explanatory power was found in the infrared band (Table 2),likely the focus was on identifying rock outcrops. The spectral reflectance of rock is typically differentiated from soil and vegetation in the infrared, which reflects moisture, mineralogy, and structure. This is important to keep in mind if the model is applied to different rock types outside of the Sierras. This also indicates that all 4 bands are important, even if other ancillary data is unnecessary.

The low resource requirements and relatively quick processing time allows for low cost experimentation while providing a powerful new analytic tool for practitioners. The main advantage we found for using CNN is that it enables researchers to build a hierarchy of local and sparse robust features extracted from spectral inputs automatically during the training process. CNN also performed better (0.95 $F1_{Macro}$) than the classical pixel-based and object-based SVM and RF models that we tested, which had 0.78–0.84 $F1_{Macro}$ scores (Table 4). The object-based (OBIA) model (0.82–0.84 $F1_{Macro}$) performed better than the pixel-based model (0.78–0.80 $F1_{Macro}$), but the CNN model was still a statistically significant improvement ($p < 0.001$) over using OBIA. Though we only trained the classical models using two test sites (ELD and ELD2), the sites used have the least spectral ambiguity (i.e., predominantly bright granite bedrock or dark green forest) making them the easiest sites to accurately classify and thus to generate high $F1_{Macro}$ scores.

For CNN, the mean results were calculated using subsets of all eight test areas in the Sierras, which were carefully chosen to represent a range of bedrock (lithology type, color, and texture) and vegetation covers (forest, grasslands, etc.) present across the Sierras. The high accuracy we achieved was dependent on the quality of the input data (both mapping and imagery), training technique, and algorithm performance. As shown in Table 5, the $F1_{Macro}$ measure of accuracy was lowest for the CAL (0.93), PLA (0.92), and MAR (0.93) sites, but these are still higher than most previous studies. These slightly lower values likely reflect the challenges in differentiating spectrally similar pixels (e.g., dry grass vs. tan bedrock in CAL and PLA, and loose rock (talus) vs. intact bedrock in MAR). However, by training the model across a well-chosen spectrum of test areas with the additional label refinement, the CNN model was able to perform well overall and is hopefully robust enough to perform well over the entire Sierras. Compared to the initial mapping, the CNN model actually does a better job of separating trees from rocks Figure 12 and limits mixed areas of rocks and trees. The accuracy was lowest in the areas at the intersection of rock and non-rock surfaces. However, this is an acceptable error that does not detract from locating rock vs. non-rock occurrences spatially.

Despite the fact that we used a generalizable, deep-learning approach—combining feature extraction and classification in one trainable model on a pixel-by-pixel basis—it is still possible to interpret features. Feature maps extracted from the first convolutional layer (e.g., Figure 13) provide a better understanding of which features are automatically learned during the training process. These feature maps suggests that the proposed convolutional neural network has learned to recognize patterns related to classification classes, making this a more promising approach compared to the classical methods.

Calculating the model accuracy over a wider area is unfortunately not possible because of the over- or underestimation problem inherent with the existing surface cover maps in the area (e.g., NLCD). With the exception of manual comparison, there is no readily available data to test the accuracy of our results outside these mapped test ares. However, by training with these eight sites, chosen to capture the variability across the Sierras, the accuracy is unlikely to decrease significantly if run over the entire Sierras (e.g., the region outlined in Figure 2). The Sierras are an ideal test ground because of their large swaths of exposed rock and the stark contrast between the light-colored rocks and the darker vegetation. Testing over the entire Sierras would be the next step in verifying the feasibility of extending this workflow over an even larger area (e.g., statewide) while maintaining similar accuracy.

This model could possibly be further improved in the future with DEM data (e.g., slope, elevation, aspect) or with NDVI to break the other class into more precise surface covers (e.g., water and vegetation types). Additionally, we could conduct testing with lower spatial resolution data that has higher spectral resolution, e.g., Landsat [27], to provide more coverage in the infrared range, or AVIRIS (where available) [64], to provide higher temporal resolution (e.g., daily to monthly imagery) to reduce noise from seasonal changes in vegetation and soil.

## 6. Conclusions

In this paper, we simplified the surface cover classification problem to differentiating exposed bedrock (or *rock*) from "not rock" (or *other*) in the Sierras, and we explored how to improve classification techniques using machine learning and widely available imagery. A combination of manual and semi-automated mapping was used to classify initial data, which was refined using manual and automated techniques. A variety of machine learning models were explored, and we found that the CNN model best matches the test area mapping. The CNN model proposed in this paper demonstrates high overall accuracy (up to 0.95 $F_1$ score) using eight diverse test sites in the Sierras and publicly available high-resolution 4-band NAIP imagery to identify rock outcrops versus other surface covers using standard computing hardware. We found that the near-infrared band can be useful for model training without need for ancillary DEM data such as texture, slope, or elevation. The approach could in future be scaled up to larger areas with sufficient high-quality imagery and training sets informed by geologic mapping. This workflow showed that, with minimal training, it is feasible to accurately identify surface exposure of bedrock over diverse test sites across the Sierras. This is promising for scaling up this model for larger areas, like the entire Sierra Nevada mountains. Improving rock classification for surface cover maps is potentially of use to resource managers, geologists, and environmental stewards alike. With abundant, high-quality imagery increasingly available worldwide, we show that the challenge of turning it into meaningful Earth surface data can begin to be addressed by well-developed machine learning models whose workflows leverage high-quality training data that is informed by geologic, topographic, or other surface maps. This is just one example, where the successful machine learning approach benefits from iterative expert input on both the mapping and programming in order to address a surface cover classification challenge.

**Author Contributions:** Conceptualization, J.S., V.Z. and C.C.-D.; methodology, H.P. and C.C.-D.; software, H.P.; validation, H.P. and C.C.-D.; formal analysis, H.P.; investigation, H.P. and C.C.-D.; resources, J.S. and V.Z.; data curation, H.P.; writing—original draft preparation, H.P. and C.C.-D.; writing—review and editing, H.P. and C.C.-D.; visualization, H.P. and C.C.-D.; supervision, J.S. and V.Z.; project administration, J.S. and V.Z.; funding acquisition, J.S. and V.Z.

**Funding:** This research was funded by a Creative Research and Development Agreement (CRADA) between USGS and Digamma.ai, executed in 2017 for the purposes of exploring how AI methods can inform geologic hazard mapping. C.C.D. and J.S. received additional funding from the National Geospatial-Intelligence Agency (NGA) Military Interdepartmental Purchase Request (MIPR) number NDCVG88239GS01 and the California Department of Water Resources grant number 4600011144.

**Acknowledgments:** The authors would like to thank the three anonymous reviewers and the two USGS internal reviewers for greatly improving this paper, as well as Lizz Caplan for proofreading. The authors would also like to thank the Center for Western Weather and Water Extremes at Scripps Institution of Oceanography (UCSD) for their support and California Department of Water Resources for additional funding.

**Conflicts of Interest:** The authors declare no conflict of interest.

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
