# Peer review of "Where’s the Rock: Using Convolutional Neural Networks to Improve Land Cover Classification"

_remotesensing, doi:10.3390/rs11192211_

Round 1

Reviewer 1 Report

1.       The innovation point is unclear. Although the title is called as “geology-trained”, what is meaning? As far as I know, there are similar research, e.g. CNN-based classification, SVM, RF, etc. What is your contribution in manuscript?

2.       It is better for the content of “2.2. Potential classification models” and “2.1. Data and limitations” to move to the “1. Introduction to the Opportunity”, or add a part of “Related work”. The part of “2. Methods” should mainly describe the proposed method.

3.       In “2.5.Label refinement using KNN-matting”, how can you determine the sparse manual labels? As we know, different sparse manual label has different effects on model training, please discuss the details.

4.       The part of “2.7. Initial models based on hand-crafted features” should be moved to the “4. Results”, because the content of 2.7 is just the comparison in experiment, not the main body of method.

5.       It is confusing for the structure of the manuscript; the part of method and the part of experiment results are unclear and confused. E.g. the experiment results in Fig.4 appears in the part of method, etc.

6.       The comparison in results in unconvincing. RF and SVM are very traditional and old method, although your method is better than RF and SVM, it can not verify your advantage of your method.

7.       The experiment and result are insufficient, more exploration should be done, e.g. error analysis, parameters effect, comparison with state-of-the-art methods, etc.

Author Response

Hi,
Thank you for your review and useful tips for improving the paper.

Response to Reviewers (in bold)

Review Report 1

English language and style

( ) Extensive editing of English language and style required

( ) Moderate English changes required

(x) English language and style are fine/minor spell check required

( ) I don't feel qualified to judge about the English language and style

Comments and Suggestions for Authors

The innovation point is unclear. Although the title is called as “geology-trained”, what is meaning? As far as I know, there are similar research, e.g. CNN-based classification, SVM, RF, etc. What is your contribution in manuscript?

-       We focused on building a simple workflow to perform land-cover classification, without additional data beyond the NAIP imagery. In particular we focused on bare-rock classification, which is often omitted in land-cover classification. To do this we needed accurate training data that accounted for different rock types and differentiated between bedrock, loose rock, and soil. The final model workflow includes data labeling using KNN-matting technique, which significantly decreases the number of resources needed to label the data, and simple CNN architecture, that achieved sufficient accuracy comparable to deeper networks. We tried to clarify this in the introduction.

-       We removed “geology-trained” from the title because we agreed it was confusing shorthand. But we tried to re-iterate our hypothesis that rock classification could be improved by coupling geologist-mapped training data with computer programmer-trained model—making the case for geologist and programmer collaborations.

It is better for the content of “2.2. Potential classification models” and “2.1. Data and limitations” to move to the “1. Introduction to the Opportunity”, or add a part of “Related work”. The part of “2. Methods” should mainly describe the proposed method.

-       Done

In “2.5.Label refinement using KNN-matting”, how can you determine the sparse manual labels? As we know, different sparse manual label has different effects on model training, please discuss the details.

-       Added explanation in section 2.5

The part of “2.7. Initial models based on hand-crafted features” should be moved to the “4. Results”, because the content of 2.7 is just the comparison in experiment, not the main body of method.

-       Done

It is confusing for the structure of the manuscript; the part of method and the part of experiment results are unclear and confused. E.g. the experiment results in Fig.4 appears in the part of method, etc.

- Edited and adjusted to make clearer.

The comparison in results in unconvincing. RF and SVM are very traditional and old method, although your method is better than RF and SVM, it can not verify your advantage of your method.

-We explored patch-based and OBIA approaches extracted low-level and high-level features according to the methods proposed in the paper (https://www.mdpi.com/2072-4292/11/12/1409). More advanced methods require ancillary data such as high resolution DEM data that is not necessarily publicly available. We wanted to see if we could improve rock and soil identification using only widely available public imagery.

The experiment and result are insufficient, more exploration should be done, e.g. error analysis, parameters effect, comparison with state-of-the-art methods, etc.

- Added additional error analysis and citations of recent papers.

Thanks

Reviewer 2 Report

The authors in this manuscript (remotesensing-558187) proposed a simple convolutional neural network (CNN) model to distinguish bare rock from soil cover in Sierras. They applied the proposed method on sub-meter-scale aerial imagery from USDA’s 2016 NAIP collection. The manuscript needs English proof-reading because there are several grammatical issues in there. Overall, the work is meaningful but needs some corrections. I do have some comments and suggestions for this manuscript. Here is my specific comments with regard to this manuscript: - Abstract: The abstract should be rewritten since in the current version, its half has been covered by introduction info. Author should add some explanation about their result and how much improvements was observed by applying CNN model compared to conventional machine learning approaches.
- Introduction: In the introduction, authors should add some explanation about two types of CNN models that are commonly used in remote sensing application: patch-based vs Fully convolutional. Add information about training time and advantages of each method. They can use these papers to address it:
o Convolutional neural networks for large-scale remote-sensing image classification. IEEE TGRS, 2016.
o Very deep convolutional neural networks for complex land cover mapping using multispectral remote sensing imagery, Remote Sensing journal, 2018.
- In this manuscript, authors mostly used active sentences such as “We are approaching a time when we can image most of Earth’s surface daily.” That is not much common in academic writing. Please consider to revise most of them in passive form in the revised version.
- Line 53: It is not common to start a sentence with but in academic writing.
- Figure 3: Author should add North and scale bar.
- Conclusion: Add the result of comparison of ML methods and the improvement of CNN approach. Also, add suggestions for future work at the end of conclusion.

Author Response

Hi,
Thank you for your review and useful tips for improving the paper.

Response to Reviewers (in bold)

Review Report 2

English language and style

(x) Extensive editing of English language and style required

( ) Moderate English changes required

( ) English language and style are fine/minor spell check required

( ) I don't feel qualified to judge about the English language and style

Comments and Suggestions for Authors

The authors in this manuscript (remotesensing-558187) proposed a simple convolutional neural network (CNN) model to distinguish bare rock from soil cover in Sierras. They applied the proposed method on sub-meter-scale aerial imagery from USDA’s 2016 NAIP collection. The manuscript needs English proof-reading because there are several grammatical issues in there. Overall, the work is meaningful but needs some corrections. I do have some comments and suggestions for this manuscript. Here is my specific comments with regard to this manuscript: - Abstract: The abstract should be rewritten since in the current version, its half has been covered by introduction info. Author should add some explanation about their result and how much improvements was observed by applying CNN model compared to conventional machine learning approaches.

- Introduction: In the introduction, authors should add some explanation about two types of CNN models that are commonly used in remote sensing application: patch-based vs Fully convolutional. Add information about training time and advantages of each method. They can use these papers to address it:

o Convolutional neural networks for large-scale remote-sensing image classification. IEEE TGRS, 2016.

o Very deep convolutional neural networks for complex land cover mapping using multispectral remote sensing imagery, Remote Sensing journal, 2018.

-       Re-wrote the abstract, added an explanation of the two types of CNN models to section 2.2, and cited the suggested paper.

- In this manuscript, authors mostly used active sentences such as “We are approaching a time when we can image most of Earth’s surface daily.” That is not much common in academic writing. Please consider to revise most of them in passive form in the revised version.

- Line 53: It is not common to start a sentence with but in academic writing.

- Figure 3: Author should add North and scale bar.

 - Done

- Conclusion: Add the result of comparison of ML methods and the improvement of CNN approach. Also, add suggestions for future work at the end of conclusion.

- Done

Reviewer 3 Report

This article reports on the mapping of rock from fine spatial resolution optical imagery. This is a potentially interesting topic that may be of interest to the journal’s readership. There are some issues that may require attention:

1.      The article is based on the use of machine learning approaches that are very well known and established. There is nothing particularly new or novel in the methods – although it is an interesting and worthy case study of contemporary methods. To make the article stronger could aspects be enhanced – e.g. how may carbon models be enhanced?

2.      Are the models of carbon etc. spatially explicit? If not, do you actually need highly accurate maps or can you cope with less accurate maps that allow extremely accurate area estimation? The introduction talks of problems of over- and under-estimation – if the errors are known (as they would be from a properly validated mapping exercise) then the area estimates can be adjusted for mis-classification error.

3.      The article over-states itself. The text is written in a way that makes the methods sound completely generalizable etc. yet the work is clearly not. The region of study is large but the complexity limited with a focus on broad lithological classes. There is also no reason to over-state the work, it is an interesting regional scale study that demonstrates the potential of a method that (as any machine leaning method would be) if trained appropriately would be scalable.

4.      Some of the text is over bold – such as that below Table 4 which includes very bold statements without a shred of evidence or support.

5.      While some text could easily be trimmed (e.g. parts of the introduction that cover very well-known material) some needs expansion. For example there is a need for more information on the segmentation (line 260) and the KNN-matting (section 2.5); the latter is important as this is one of the more interesting parts of the results presented.

6.      There is no surprise in the results. Indeed the results need strengthening. Are there not more results that could be shown?  Also it is insufficient simply to compare the magnitude of accuracy estimates to then claim that one approach is better than another (e.g. text linked to Table 5). These are accuracy estimates and need to be compared in a statistically rigorous fashion. Are the differences statistically significant? Also how useful is a comparison of OA when the classes so imbalanced? More rigorous analysis is required and surely the attention should be on the accuracy of rock classification? As it stands Table 5 could actually be masking terrible performance as the focus is on OA not the specific feature of interest. It would be perfectly possible for the accuracy of rock classification to be declining – I do not believe this to be the case but the authors need to provide clear evidence to support their case.

Overall – this article over-sells itself but is still a potentially interesting contribution on the value of contemporary image analysis methods for an interesting application. The work needs some revision, notably to the rigour of the accuracy assessments and comparison.

Author Response

Hi, 
thanks for the review. 

Response to Reviewers (in bold)

Review Report 3

English language and style

( ) Extensive editing of English language and style required

( ) Moderate English changes required

(x) English language and style are fine/minor spell check required

( ) I don't feel qualified to judge about the English language and style

Comments and Suggestions for Authors

This article reports on the mapping of rock from fine spatial resolution optical imagery. This is a potentially interesting topic that may be of interest to the journal’s readership. There are some issues that may require attention:

The article is based on the use of machine learning approaches that are very well known and established. There is nothing particularly new or novel in the methods – although it is an interesting and worthy case study of contemporary methods. To make the article stronger could aspects be enhanced – e.g. how may carbon models be enhanced?

      -Added how this could be used for landslide susceptibility model, and re-iterated this method focused on rock identification which has large errors in current land cover models.

Are the models of carbon etc. spatially explicit? If not, do you actually need highly accurate maps or can you cope with less accurate maps that allow extremely accurate area estimation? The introduction talks of problems of over- and under-estimation – if the errors are known (as they would be from a properly validated mapping exercise) then the area estimates can be adjusted for mis-classification error.

      - Some models that would benefit from this information are spatially explicit (like the landslide hazard example we added). But yes, this would be an appropriate way to approach the errors for models that just need the overall area.

The article over-states itself. The text is written in a way that makes the methods sound completely generalizable etc. yet the work is clearly not. The region of study is large but the complexity limited with a focus on broad lithological classes. There is also no reason to over-state the work, it is an interesting regional scale study that demonstrates the potential of a method that (as any machine leaning method would be) if trained appropriately would be scalable.

-We added language to dampen our assertion that this would work for the entire Sierras with the same accuracy. We also re-iterated that we trained the model with 8 sites chosen to represent the range of lithology, vegetation, etc. expected in the wider area.

Some of the text is over bold – such as that below Table 4 which includes very bold statements without a shred of evidence or support.

-Added explanation and additional citation.

While some text could easily be trimmed (e.g. parts of the introduction that cover very well-known material) some needs expansion. For example there is a need for more information on the segmentation (line 260) and the KNN-matting (section 2.5); the latter is important as this is one of the more interesting parts of the results presented.

- Done

There is no surprise in the results. Indeed the results need strengthening. Are there not more results that could be shown? Also it is insufficient simply to compare the magnitude of accuracy estimates to then claim that one approach is better than another (e.g. text linked to Table 5). These are accuracy estimates and need to be compared in a statistically rigorous fashion. Are the differences statistically significant? Also how useful is a comparison of OA when the classes so imbalanced? More rigorous analysis is required and surely the attention should be on the accuracy of rock classification? As it stands Table 5 could actually be masking terrible performance as the focus is on OA not the specific feature of interest. It would be perfectly possible for the accuracy of rock classification to be declining – I do not believe this to be the case but the authors need to provide clear evidence to support their case.

-To strengthen the results we calculated Kappa index and F1 score and added McNemar test.

- We also used confusion matrices, which gives the accuracy for each test site of identifying what was originally mapped as rock (via true/false positives and negatives).

Overall – this article over-sells itself but is still a potentially interesting contribution on the value of contemporary image analysis methods for an interesting application. The work needs some revision, notably to the rigour of the accuracy assessments and comparison.

Round 2

Reviewer 1 Report

The manuscript has been improved a lot, but there is a little issue that is necessary to be solved. The conclusions to the related work are not sufficient, e.g.

Land Cover Classification from fused DSM and UAV Images Using Convolutional Neural Networks. Remote Sensing, 2018, 11(12), 1461. Urban land use and land cover classification using novel deep learning model based on high spatial resolution satellite imagery. Sensors. 2018, 18(11): 3717. Object-Based Land Cover Classification of Cork Oak Woodlands using UAV Imagery and Orfeo ToolBox. Remote Sensing, 2018, 11(10), 1238.

Etc. Especially the work in recent 3 years. And compare the difference and similarity between your work and related work.

Author Response

Thank you for your comments.

We re-wrote section 2.2 and cited the suggested papers or other similar papers in lines 133-145. We highlighted the differences and similarities between the related work and our work, and revisited this briefly in the discussion..

We understand that our approach is well known. The additional contribution we made was implementing it with a manually trained inventory, and improving the workflow to more accurately differentiate rock from soil for land cover classification.

Reviewer 2 Report

The authors did not apply all of my comments in the previous revision, e.g., adding north sigh to figures. The manuscript still has several writing problems and needs to be largely reworked and proofread. They should also provide a cover letter to answer all reviewers comments one by one. Accordingly, this manuscript can be considered for publication after major revision.

Author Response

We apologize that our previous line-by-line response and edits were insufficient or misconstrued, and thank you for taking the time to review this again. We had attempted to address all your comments. We will reiterate our changes below (bold), and elaborate (in italics). In addition, we have re-read the manuscript and given the manuscript to an english-speaking proofreader. 

Previous Comments and Suggestions for Authors (reponse in bold, elaboration in italics)

The authors in this manuscript (remotesensing-558187) proposed a simple convolutional neural network (CNN) model to distinguish bare rock from soil cover in Sierras. They applied the proposed method on sub-meter-scale aerial imagery from USDA’s 2016 NAIP collection. The manuscript needs English proof-reading because there are several grammatical issues in there. Overall, the work is meaningful but needs some corrections. I do have some comments and suggestions for this manuscript. Here is my specific comments with regard to this manuscript: - Abstract: The abstract should be rewritten since in the current version, its half has been covered by introduction info. Author should add some explanation about their result and how much improvements was observed by applying CNN model compared to conventional machine learning approaches.

- Introduction: In the introduction, authors should add some explanation about two types of CNN models that are commonly used in remote sensing application: patch-based vs Fully convolutional. Add information about training time and advantages of each method. They can use these papers to address it:

o Convolutional neural networks for large-scale remote-sensing image classification. IEEE TGRS, 2016.

o Very deep convolutional neural networks for complex land cover mapping using multispectral remote sensing imagery, Remote Sensing journal, 2018.

- Re-wrote the abstract, added an explanation of the two types of CNN models to section 2.2, and cited the suggested paper plus additional references. We have also added additional citations and expanded our description of recent work on land cover classification using neural networks to this version of the manuscript in response to reviewer comments (lines 133-145 in particular).

- In this manuscript, authors mostly used active sentences such as “We are approaching a time when we can image most of Earth’s surface daily.” That is not much common in academic writing. Please consider to revise most of them in passive form in the revised version.

- Line 53: It is not common to start a sentence with but in academic writing.

- Figure 3: Author should add North and scale bar.

- Done. We previously added north arrows and scale bars to all the images in Fig. 3. We have also attempted to address all the uncommon sentence phrasing, and had a professional english language proofreader read through this revision.

- Conclusion: Add the result of comparison of ML methods and the improvement of CNN approach. Also, add suggestions for future work at the end of conclusion.

- Done. We clarified what the improvement was and have framed this work as the basis for running a model for the entire Sierras (i.e. potential future work).

Please see the attachment to find pdf document with differences. 

Reviewer 3 Report

The authors have enhanced their article in the light of the review comments. However, concerns remain.

The style of writing is a little awkward and this has a negative impact on the article.

The article still includes inappropriate text and some errors. As just a few examples:

-          The very first sentence of the introduction is untrue. Daily images of the Earth have been acquired for decades.

-          Text under table 4 still contains assertions without the slightest shred of evidence offered in support. I would certainly argue their validity. Where is the evidence to support the statement “Hand-crafted features are usually not robust and are computationally intensive due to high dimensions. The discriminative power is usually low.” The authors seem to be over-stating the situation especially as the methods assessed need not have been optimised fully.

-          Text in Table 1 is untrue – some handcrafted approaches will require very large training sets.

-          While the authors use F1 instead of OA they should note that this in no way helps with the class imbalance issue.  Like OA, the F1 measure is very sensitive to class imbalance. This is very well known and arises especially because precision is highly sensitive to imbalance.

-          The confusion matrices are poorly presented. In addition the provision of information as % is unhelpful.

Overall – the article has been improved but concerns remain.

Author Response

The style of writing is a little awkward and this has a negative impact on the article.

-Thanks for taking the time to review this again. We tried to address the awkward writing by having an english-language proof-reader review our revised manuscript prior to resubmission.

The article still includes inappropriate text and some errors. As just a few examples:

-          The very first sentence of the introduction is untrue. Daily images of the Earth have been acquired for decades.

-It was not our intention to imply that daily imagery is a new thing. We changed the sentence from “With daily Earth surface imagery, it is easy to find orthorectified imagery for almost every location globally, at meter-scale resolution.” to “An increasing abundance of publicly-available Earth surface imagery makes it easy to find orthorectified imagery for nearly every location in the world, often at meter-scale or better resolution.”

-          Text under table 4 still contains assertions without the slightest shred of evidence offered in support. I would certainly argue their validity. Where is the evidence to support the statement “Hand-crafted features are usually not robust and are computationally intensive due to high dimensions. The discriminative power is usually low.” The authors seem to be over-stating the situation especially as the methods assessed need not have been optimised fully.

-We clarified and simplified the statement to not over-state the comparison we are trying to make between the classical approaches vs. CNN pros and cons (lines 336-341).

“All methods implemented for extracting spatial features generated only low-level features requiring empirical parameters (e.g., neighbor size). Spatial features, therefore, depend on expert knowledge and parameter setting, which is why it is difficult to find universal parameters to extract appropriate features for each type of land cover surface using these methods. Consequently, spatial features are usually not robust and have poor generalizability. ” 

-          Text in Table 1 is untrue – some handcrafted approaches will require very large training sets.

- We agree that we couldn’t adequately simplify the difference in a table, so we removed Table 1. Instead we added text explanation in section 2.2 (lines 145-157) . 

“The above studies show that classification models based on hand-created features are easily interpreted and assume that the extracted features are robust to the variances in the data but need to be manually engineered. Deep learning-based models, like CNN, present a generalized approach using feature extraction and classification in one trainable model. However, analyzing and visualizing the features is difficult. The variety of methods used in recent studies shows there is no consensus yet on the best model for solving land cover classification problems.”  

-          While the authors use F1 instead of OA they should note that this in no way helps with the class imbalance issue.  Like OA, the F1 measure is very sensitive to class imbalance. This is very well known and arises especially because precision is highly sensitive to imbalance.

- To address this concern, we added the Macro F1 score. In macro-averaging, each metric is averaged over all classes. Thus, if the F1 for the ‘rock’ class were low, it would decrease final F1 score. Therefore these scores should represent the performance of the model independent of the class imbalances. We added a citation and explanation of this method in the results section.

-          The confusion matrices are poorly presented. In addition the provision of information as % is unhelpful.

- We redid the confusion matrix so the location of the actual vs. predicted results are clearer. We also added F1 score per site.

Overall – the article has been improved but concerns remain.

-----------------

Please see the attachment to find pdf document with differences. 

Round 3

Reviewer 2 Report

I'm satisfied that the authors have addressed my comments now, and I consider the paper ready for publication.

Author Response

Thank you for continuing to review this paper, and for the constructive feedback. We agree it is just one small step forward in geologic applications, but still an improvement. We are glad you find it promising. We look forward to expanding this effort in the future to test the accuracy across the Sierras and to see if the model continues to perform well in other "barren lands".

We have completed the additional minor revisions per the editor and reviewer recommendations. In particular:
1) We have proof-read and spell-checked the manuscript again and hope to have caught the remaining typos and awkward phrasing. We also changed the last sentence in the abstract as suggested by the editors.
2) We made a new methods subsection called "Accuracy assessment" and moved the description of statistical methods here. We also combined the two F1/Kappa tables into a single table in the results section (i.e. the former Table 3 was removed and combined with the former Table 5).
3) We attempted to refine the sweeping statements during the final proof-reading.
4) We added additional citations and elaborated on our claims throughout the paper (i.e. differences in models, statistical methods used, etc.).
5) We included peer-reviewed articles along with the grey literature where applicable (to cite the original work as well as the peer-reviewed use of the models, datasets, etc.).

Reviewer 3 Report

The article has been enhanced. I still have concerns (e.g. the accuracy measures are still highly sensitive to class imbalance) but these feel too detailed to dwell on.

Author Response

(The authors gave the same response as above.)
